# Monitoring Response to Home Parenteral Nutrition in Adult Cancer Patients

**DOI:** 10.3390/healthcare8020183

**Published:** 2020-06-23

**Authors:** Paolo Cotogni, Riccardo Caccialanza, Paolo Pedrazzoli, Federico Bozzetti, Antonella De Francesco

**Affiliations:** 1Pain Management and Palliative Care, Department of Anesthesia, Intensive Care and Emergency, Molinette Hospital, University of Turin, 10126 Turin, Italy; 2Clinical Nutrition and Dietetics Unit, Fondazione IRCCS Policlinico San Matteo, 27100 Pavia, Italy; r.caccialanza@smatteo.pv.it; 3Medical Oncology Fondazione IRCCS Policlinico San Matteo and Department of Internal Medicine and Medical Therapy, Università degli Studi di Pavia, 27100 Pavia, Italy; p.pedrazzoli@smatteo.pv.it; 4Faculty of Medicine, University of Milan, 20122 Milan, Italy; federicobozzetti@gmail.com; 5Clinical Nutrition, Department of Internal Medicine, Molinette Hospital, 10126 Turin, Italy; adefrancesco@cittadellasalute.to.it

**Keywords:** oncology, nutritional status, nutritional support, artificial nutrition, home care, guidelines, clinical practice

## Abstract

Current guidelines recommend home parenteral nutrition (HPN) for cancer patients with chronic deficiencies of dietary intake or absorption when enteral nutrition is not adequate or feasible in suitable patients. HPN has been shown to slow down progressive weight loss and improve nutritional status, but limited information is available on the monitoring practice of cancer patients on HPN. Clinical management of these patients based only on nutritional status is incomplete. Moreover, some commonly used clinical parameters to monitor patients (weight loss, body weight, body mass index, and oral food intake) do not accurately reflect patient’s body composition, while bioelectrical impedance analysis (BIA) is a validated tool to properly assess nutritional status on a regular basis. Therefore, patient’s monitoring should rely on other affordable indicators such as Karnofsky Performance Status (KPS) and modified Glasgow Prognostic Score (mGPS) to also assess patient’s functional status and prognosis. Finally, catheter-related complications and quality of life represent crucial issues to be monitored over time. The purpose of this narrative review is to describe the role and relevance of monitoring cancer patients on HPN, regardless of whether they are receiving anticancer treatments. These practical tips may be clinically useful to better guide healthcare providers in the nutritional care of these patients.

## 1. Definition

Home parenteral nutrition (HPN) refers to the administration of nutritional support through a central venous access device (CVAD) at home [1].

## 2. Indications

It is well known that cancer patients on chemo- or radiation therapy may experience adverse events (toxicity) related to these treatments [2]. Frequently, these adverse effects cause nutrition-impact symptoms (nausea, vomiting, anorexia, early satiety, diarrhea, constipation, dry mouth, mouth sores, problems with smell, taste, or swallowing, fatigue and pain) that impair the patient’s food intake. Additionally, toxicity often worsens the decrease in oral food intake already present in cancer patients due to the disease [3]. These patients need nutritional assessment and, if they are at risk of malnutrition or malnourished, guidelines strongly recommend a nutritional intervention [4,5,6]. Specifically, in patients undergoing anticancer treatments, if nutrients intake remains inadequate despite skilled counseling and oral nutritional supplements (ONS), guidelines recommend that supplemental enteral nutrition (EN) or, if this is not adequate or feasible, parenteral nutrition (PN) should be prescribed. Additionally, in patients with persistent deficient nutrients intake, home artificial nutrition (either EN or PN) in suitable patients is strongly recommended. Although provision of HPN also delivers nutrients to cancer cells, there is no evidence-based data that this possibility has harmful impact on survival. Consequently, this hypothetical risk should have no effect on the decision to feed a cancer patient when HPN is clinically indicated [4].

Because tailoring nutritional support to individual needs is also beneficial in advanced cancer patients receiving no chemotherapy, guidelines recommend the assessment of nutritional deficiencies in all of these patients [4,5,6]. This is the case of incurable cancer patients with chronic oral food or enteral feeding intolerance (abdominal pain, untreatable nausea or vomiting, malabsorption or diarrhea) or with chronic intestinal failure due to malignant inoperable bowel obstruction (intra-abdominal recurrences and/or peritoneal carcinomatosis), short bowel syndrome, radiation enteritis, and high-output fistula or ileostomy [7].

In incurable patients who are aphagic or severely hypophagic, the indication for HPN is controversial. However, guidelines state that HPN is not contraindicated if anticancer treatment has been discontinued [7]. In patients with incurable cancer and a short life-expectancy (less than two months), HPN is not recommended, while it should be administered if the patient is likely to die earlier because of malnutrition rather than the progression of cancer disease [4,7].

## 3. Prevalence

An extensive disparity can be found in data reporting the primary diagnostic category of patients who require HPN [8]. In many countries, the most frequent medical condition of HPN receivers is cancer disease; however, the prevalence of HPN varies around the world due to different organizational structures, legislations, and healthcare policies, and also due to medical, economic, cultural, religious, and ethical features. For instance, prescribing HPN in patients with cancer is not a common practice in UK, The Netherlands, and Denmark [9].

The first experiences of patients receiving PN at home began to be reported in the late 1960s and early 1970s. Nowadays, around 25,000 patients are estimated to be on HPN in the United States [10]. In 2012, the primary medical condition for HPN in adults in Italy was oncological (60%); specifically, point prevalence (number of cases/million inhabitants) of HPN was 15.4 for adult cancer patients [11]. In European countries, the only survey concerning the practice of artificial nutrition which could be compared with the Italian survey is a report by the British Association for Parenteral and Enteral Nutrition (BAPEN) carried out in 2010. A comparative analysis shows that the HPN prevalence was nearly 2.5-fold greater in Italy, while the practice of HPN in adult cancer patients was only 10–20% in the UK. This great difference may be related to the attitude towards the use of this therapy in Italy in oncology patients [11].

Although several observational studies suggest the benefit of nutritional support in patients receiving chemo- or radiation therapy [12], the literature shows that it has not been adequately considered by oncologists, yet [13,14,15,16,17,18,19]. In these articles, the authors identify several factors acting as barriers to early use of nutritional treatment in cancer patients. Particularly, they underline the lack of knowledge and/or training in the decision-making for nutritional pathway (assessment, provision, and monitoring).

## 4. Transition from Hospital to Home Care

Consistent consideration should be used in managing the transition of patients who are candidates for HPN from hospital to home care [20]. This issue is really relevant as patient’s safety always takes priority over attaining nutritional goals. Indeed, patients who face PN at home may be initially worried by the management of this therapy [21]. Therefore, the patient and/or his/her caregiver(s) should be trained to self-sufficiently perform procedures related to the infusion of the HPN bag and the manipulation of the CVAD. Indeed, prevention of HPN-related complications should be the main effort when discharging home patients on HPN. However, patient and/or caregiver training prior to discharge—or in the outpatient setting—is limited. Therefore, the education process should be continued at home by skilled nurses. Additionally, the training should include procedures for self-monitoring (e.g., weight and edema) and recognition of possible complications (e.g., CVAD mechanical complications; fever that occurs at the infusion start is a potential sign of bacterial colonization of the catheter). Finally, nursing visits at home, follow-up in-hospital appointments, and laboratory monitoring should be scheduled as well as patients/caregivers should be supplied with written instructions on when and who to call if complications occur.

## 5. Administration of HPN

PN can be total (TPN) when patients have no or negligible oral/enteral nutrition (<200 kcal/day) [22] or supplemental (SPN) when PN is added to oral/enteral nutrition to achieve protein and energy requirements [23].

Generally, SPN at home provides 1000–1250 kcal/day from three to six times per week in patients with residual—but insufficient—oral food intake. In comparison with TPN, SPN has a low risk of both overfeeding and refeeding syndrome (RS), as well as hyperglycemia, overhydration, and liver dysfunction.

Specifically, RS is defined as the potentially fatal shifts in fluids and electrolytes that may occur in severely malnourished patients at the start of artificial nutrition (either EN or PN). The risk of onset of RS increases with the degree of the patient’s malnutrition. Guidelines recommend, if oral food intake has been severely decreased for a long period, to increase nutrition slowly over several days to prevent RS [6].

SPN also offers benefits on the quality of life (QoL) because it reduces the number of nights potentially at risk of sleep disturbance. Finally, reducing the number of infusion days SPN lowers manipulation of the CVAD and therefore the risk of catheter-related complications. In recent studies, SPN was the preferred approach of provision of PN in oncology outpatients [24,25].

## 6. Why Is Monitoring Important?

The development and implementation of healthcare actions that are viable, affordable, and successful in the hospital setting is a challenge [26]. Nowadays, no strong recommendations are available to suggest the use of definite indicators or the timing of tools used to monitor the response to HPN in cancer patients [8]. Moreover, the monitoring process may be less easy to perform in outpatients than inpatients and this issue is to be faced in the set-up of the home nutritional service [6].

The HPN monitoring process aims at determining the appropriateness of nutritional therapy, ensuring the achievement of nutritional goals, and reducing the risk of complications [8]. Additionally, a consistent close monitoring for HPN-related complications is able in reducing unplanned hospital admissions and overall associated costs [20].

In 2011, a board of experts in the management of adult cancer patients on HPN developed a list of interventions related to quality of care. These experts indicated relevant differences in care practices regarding cancer patients compared with benign patients [9]. Specifically, the experts considered monitoring of liver function tests, vitamins, trace elements or metabolic bone diseases not essential in oncology patients due to their expectancy of survival. Additionally, teaching and training of HPN monitoring is transferred in part or completely in the home setting in the presence of caregivers/family members.

## 7. Monitoring Process

For many years, we have used the approach described hereinafter to monitor the response to HPN in cancer patients (Figure 1) [25]. However, the practical tips described in this review are mostly based on expert opinion.

The application of these suggestions could be facilitated by implementing dedicated services and pathways, and assigning duties to dedicated healthcare providers (i.e., physicians and dietitians expert in clinical nutrition in oncology) in each oncologic center. The evaluation of eligibility of a cancer patient for the HPN program is requested mainly by oncologists, but also by surgeons, internal medicine physicians, and general practitioners. Inpatients are assessed during the consultations carried out in the wards before discharge, while outpatients are assessed in dedicated hospital rooms in the Comprehensive Cancer Center. The criteria for accepting patients in the HPN program follow the European guideline recommendations for eligibility [4] and were already described [25].

After the start of HPN, patients are monitored on a regular basis (at least every two weeks) through planned and structured telephone interviews by the clinician in charge of the HPN. Additionally, visits at home by the nurse and general practitioner are scheduled initially every day for two to three weeks and every week thereafter. Only after appropriate training can the patient’s caregiver self-manage PN. Telephone assistance by a physician is available for patients as well as their caregivers and healthcare providers at all times. HPN is delivered using standard nutritional bags, commercially manufactured and ready-to-use, containing glucose, amino acids, lipids, and electrolytes overnight for 10–14 h per day through a CVAD. The HPN program is personalized to meet calories, protein, and water requirements. Generally, HPN is prescribed to provide a range between 25 and 30 kcal per kg per day, depending on the patient’s activity of daily living, and an amino acid supply between 1 and 1.5 g per kg per day. After the start of HPN, approximately every month (±5 days), the patient undergoes a hospital re-evaluation by both the dietitian (including a 24-h food recall) and clinician. If the patient needs admission to hospital, he/she is referred to the HPN-teaching hospital.

Since 2000, we have been applying an approach characterized by the close intersection between the oncological and the nutritional pathway. This approach brings oncologists and experts in clinical nutrition to develop anticancer and nutritional therapies simultaneously during all of the course of the patient’s disease. This approach ensures that cancer patients can receive the nutritional treatments they need as early as possible and as part of standard care.

## 8. Anthropometric Measures

Body weight is surely the most common indicator used to monitor cancer patients on HPN. However, handling nutritional requirements based only on actual body weight or weight changes is confusing. Indeed, when evaluating body weight misleading factors as ascites, edema, or pleural effusion should be taken into account.

Similarly, body mass index (BMI) may be a confounding measure. Indeed, patients may have same BMI, but markedly different lean mass. For instance, an obese patient may have an important loss of lean body mass that is hidden by a significant mass of fat. This issue is relevant because sarcopenic obesity is associated with an increase in incidence of dose-limiting toxicity [2].

## 9. Assessment of Oral Food Intake

A recurrent and scheduled quantification of oral food intake is also important in patients on HPN, especially in those on SPN. Indeed, patient’s nutrient balance is determined by oral intake plus SPN support minus expenditure. Additionally, the assessment of food intake can also evaluate whether the severity of patient’s nutrition-impact symptoms is likely to improve or increase. Therefore, this assessment is a key element for monitoring patients and may predict earlier than weight loss the need of a switch from SPN to TPN.

The evaluation of patient’s oral food intake is performed by skilled dietitians using a well-designed interview (24-h food recall). During this interview, the dietitian asks the patient to report which foods and drinks he/she has eaten during the previous 24 h. However, this tool has considerable limits and potential significant errors, even if it is performed by skilled personnel (Figure 2).

## 10. Assessment of Nutritional Status

Screening is a simple and rapid process to select patients who are at risk of malnutrition or malnourished. Conversely, assessment of nutritional status is a diagnostic process, which individualizes the degree of malnutrition, and it is obviously more complex and time-consuming than screening.

The analysis of nutritional status is useful to complete objective assessments with subjective evaluations. The Patient-Generated Subjective Global Assessment (PG-SGA) [27] has demonstrated to be effective in assessing nutritional status in cancer patients [28,29]. The PG-SGA uses semi-quantitative and qualitative indicators to classify patients in category A (well-nourished), category B (moderately malnourished), or category C (severely malnourished).

This tool has five items: weight, nutrient intake, nutrition-impact symptoms, functioning, and physical exam). It was designed so that the components of the medical history can be completed by the patient using a check box format while the physical exam is performed by a physician or dietitian. History weight loss is a more accurate predictive factor than actual body weight as well as severe deficit of oral food intake and the presence of nutrition impact symptoms. The fourth item of this tool is the patient’s functional capacity or energy level (bedridden to full capacity). Finally, the PG-SGA evaluates muscle wasting (quadriceps, deltoids) and loss of subcutaneous fat by a physical examination. Inspection and palpation for edema and ascites are important because cannot only be indicators of patient’s status, but also suggests that actual body weight or weight changes may be misleading elements.

Several studies showed the usefulness of PG-SGA to evaluate patient’s nutritional status, as well as its ability to predict mortality in oncology patients [30]. Specifically, patients who were ranked as severely malnourished or declined from category B to C had the shortest median survival, and those who improved to a rating of category A had the longest median survival [31]. Therefore, by monitoring response to HPN using PG-SGA we can assume that category C is a strong predictor of mortality and category A is a predictor of longer survival.

## 11. Prognostic Indicators

Functional status is a key element to monitor in cancer patients on HPN. For assessing the performance status, both the Karnofsky Performance Status (KPS) (0–100 points) and Eastern Cooperative Oncology Group (ECOG) scale (0–5 grades) can be used. Indications that higher KPS scores increase the probability of survival in these patients has been reported in observational [22,32,33] and survival prediction studies [34]. A prospective study found that after 90 days from starting HPN, the performance status of cancer patients improved (a median increase in KPS of 10 points) [31]. In actuality, moving from a KPS of 70 to 80 means the difference in being able to carry out daily activities with no special care needed. In addition to the effect on daily living, a higher KPS has been associated with an increase in survival in oncology patients on HPN [22,35]. Recently, in a large sample of cancer patients on HPN, survival analyses were able to highlight how even small increases in KPS score (10-point increases) could significantly have a protective effect on survival [25].

Evidence shows that the presence of a systemic inflammation is associated with loss of lean body mass in cancer patients. Systemic inflammatory response is graded using scores as modified Glasgow Prognostic Score (mGPS) (0–2 grades and 2 as the worst value) (Figure 3) [36].

The prognostic value of mGPS in predicting survival in cancer patients has been demonstrated in several studies showing that mGPS was a strong predictor of survival [37,38]. In particular, an association between an mGPS score of 2 and a 160% decrease in survival was noted [38]. When measured at start of HPN, elevated mGPS scores have been associated to an increased risk of mortality [22,33]. In particular, a score of 2, is high predictive of mortality [25]. Indeed, mGPS has been included in a nomogram predicting the survival in oncology patients receiving HPN [34]. In cancer patients, a significant improvement of mGPS was observed after 90 days from the start of HPN, with a significant increase in the proportion of patients qualified as score 0 [31].

## 12. Bioelectrical Impedance Analysis

Cancer patients experience muscle loss and changes in fluid distribution with extracellular expansion and reduced intracellular water. The provision of nutrients and fluids established based only on patient’s body weight or BMI may be misleading, as these parameters do not reflect his/her body composition [39,40,41]. The analysis of body composition of cancer patients shows that it is specifically the muscle loss—independently from the fat loss—that predicts the risk of impaired performance status, surgical complications, treatment-associated toxicities, and mortality [6].

Assessment of body composition can be performed using the following techniques and parameters [6]:Anthropometry (mid upper-arm muscle area)Dual energy X-ray absorptiometry (appendicular skeletal muscle index)CT imaging (lumbar skeletal muscle index)Bioelectrical impedance analysis (BIA) (whole body fat-free mass index).

BIA is a non-invasive, validated, and cheap tool to analyze nutritional status and it is easily performed by dietitians or nurses. At the start and over time, HPN represents an efficacious method to assess and follow up modifications in body composition and can help clinicians to interpret changes in body weight more accurately [42]. The patient lies supine on a bed with legs apart and arms not touching the torso, with an empty bladder, after fasting and not receiving HPN for at least 8 h. A couple of standard electrodes are placed both on the ulnar part of the right wrist and the right medial malleolus.

Some BIA measures such as fat free mass (FFM) and phase angle (PA) were found to be associated not only with patient’s nutritional status, but also with survival in oncology patients [39,43]. Nevertheless, few studies investigated the effectiveness of BIA measures to monitor the response to HPN in cancer patients [44,45].

Moreover, BIA has shown its value as a prognostic tool for oncology patients undergoing anticancer treatments [46]. In a recent study, Cox regression analysis showed that some BIA measures were significantly associated with survival in oncology patients undergoing both HPN and chemotherapy; specifically, reactance at the start of HPN, resistance after 60 days, and phase angle after 90 days from the start of HPN [31]. Thus, longitudinal use of BIA can not only monitor response to HPN, but it may also help in tracking life expectancy, potentially contributing to a better management of global patient care.

## 13. Examination of Central Venous Access Device

For many years, the use of HPN in cancer patients receiving chemotherapy has been scarce. One of the most frequent causes was that clinicians were concerned about the risk of an increased rate of major complications (venous thrombosis and/or bloodstream infection) due to the manipulation of CVAD for HPN infusion. Some studies showed that HPN, if properly managed, is safe in oncology patients, even in those receiving chemotherapy, and its use has been associated with a low rate of catheter-related complications (CRCs) [47,48,49].

Since the most important and frequent complications of HPN result from the use of CVAD, a pivotal element of appropriate monitoring is its consistent examination. Visits at home by the nurse are scheduled initially every day for two to three weeks and every week thereafter. The care of CVAD should be provided by nurses and caregivers applying a strict policy of hand washing and environmental hygiene. Additionally, an appropriate asepsis when managing the CVAD and a strict policy for flushing the catheter with normal saline before and after use, with the pulsating “push/pause” plus positive pressure method, are adopted.

The examination of CVAD includes not only inspection, but palpation also, to look for signs of infection (e.g., tenderness and redness) of the exit site, the subcutaneous tunnel or pocket. Additionally, it is important to check for upper extremity edema. Finally, it is mandatory to review position of catheter tip on chest X-ray whenever catheter displacement is suspected.

The diagnosis of local infection and catheter-related bloodstream infection (CRBSI) are carried out according to the referring guidelines [50,51]. In the case of CRBSI, we remove the catheter and/or provide systemic antibiotic therapy—at least 10–14 days—associated with antibiotic lock therapy, according to guidelines [50]. VAD-related thrombosis is suspected in the case of local pain, edema, or other suggesting signs and later confirmed by ultrasound evaluation with or without color Doppler [52].

A prospective study of over 169,000 catheter days investigated the incidence rates of CRCs of 854 CVADs for HPN in cancer patients [53]. This study reported that the rate of total CRCs was 1.08/1000 catheter-days. Specifically, the incidence of CRBSIs was low (0.29/1000 catheter-days) as well as the rates of mechanical complications and VAD-related symptomatic thrombosis were low (0.58/1000 and 0.09/1000 catheter-days, respectively).

## 14. Quality of Life

Clinicians have to identify those cancer patients who are most likely to have benefits from HPN (i.e., improvement of nutritional status, QoL, performance status, and survival) [34,54].

The anxiety that surrounds the eating and related psychological distress cause negative repercussions for QoL of patients and caregivers [55]. Generally, studies about the use of HPN in these patients found that they had a favorable perception of the impact of HPN on their QoL. In most cases, patients relate their reduced QoL more to the incapacity to eat than to the HPN dependence [56,57]. Orrevall et al. reported the feeling of comfort and safety of both patients and families when the nutritional requirements were fulfilled through HPN [58].

Based on the patient’s tolerance, HPN is administered over a 10–14 h cycle (usually during nighttime). This administration method determines positive psychosocial effects on patient’s QoL because permits the patient to be unrestricted for the daytime activities.

Several studies have focused on QoL in cancer patients on HPN because it is a key element in these people. Bozzetti et al. found stable QoL scores up to two to three months before patient’s death [59]. Vashi et al. reported the provision of HPN was associated with improved KPS and QoL [60]. Culine et al. described a significant increase in QoL [61]. In a longitudinal study, advanced cancer patients undergoing chemo- or radiation therapy showed higher scores than at the start of HPN [62]. Moreover, even in patients not receiving treatments, HPN was shown to significantly improve some QoL items [57,62]. Sowerbutts et al. [63] reported that patients with ovarian cancer on HPN experienced a burden of treatment that did not reduce the beneficial effects of HPN. In the interviews, women told that motivation to live offset the constraints that are imposed by HPN, as well as patients and families recognized HPN as a lifesaver and were grateful for it.

A monitoring process based only on nutritional status is incomplete, as it does not investigate functional status and QoL of patients. For the evaluation of QoL, the European Organization for Research and Treatment of Cancer Quality of Life Questionnaire Core 30 (EORTC QLQ-C30) is commonly used (Figure 4); specifically, the validated Italian translation [64,65].

At the start of HPN, patients fill out the questionnaire in the presence of a dietitian or physician in case they need assistance. During the following outpatient visits, they themselves fill out the questionnaire without assistance, requiring an average of 15 min to complete it [62].

## 15. Weaning from or Withdrawal of HPN

For patients receiving HPN, one of the most important elements that should be regularly monitored is the possible need to modify the HPN program as well as to wean or withdraw this therapy. Monitoring the persistence of the indication to HPN should be performed by experts in managing this therapy. Reassessing persistent need for HPN or deciding to modify the HPN program following the evaluation of gastrointestinal function or nutritional status are good standards in all cancer patients on HPN. The recovery of or improvement in bowel function may occur more frequently in cancer patients on HPN receiving anticancer treatments. This is the case when there is a regression or resolution of nutrition-impact symptoms due anticancer treatments that had caused the decrease in oral intake.

Patients can be weaned from HPN by reducing the number of days that the therapy is infused each week (avoiding one or two nonconsecutive infusions per week). HPN can be fully discontinued when patients consistently consume at least 50–75% of their energy and protein requirements by oral route, unless intestinal failure causes severe malabsorption of nutrients. In the case of HPN discontinuation, it is important to settle a nutritional monitoring plan to guarantee a safe transition to full oral nutrition. On the contrary, incurable cancer patients with severe intestinal failure may never successfully transit off HPN or, in some cases, may have short or transitory periods of weaning from HPN.

The provision of artificial nutrition (enteral or parenteral) in the last week(s) of life is a controversial issue. In patients who are near death, artificial nutrition is unlikely to provide any benefit. However, several studies have reported that nutritional support has often been provided to oncology patients near end of life [66].

Additionally, incurable cancer patients receiving palliative care are followed up until HPN is discontinued or they die. HPN is withdrawn in case the patient gets worse (onset of severe organ dysfunction or uncontrolled symptoms; downgrading of performance status; estimated life expectancy of hours to days, and patient will [54]. In the conscious patient, withdrawing HPN should be decided with him/her [9].

## 16. Strengths and Limitations of the Review

In recent literature, there are some reviews discussing the monitoring of patients on HPN. However, most of them are not focused on the cancer patient [67,68,69]. In actuality, the latter has characteristics that distinguish it from the patient with benign chronic intestinal failure [9]. In addition, in the case of reviews regarding cancer patients, the attention devoted to monitoring is usually limited [70]. The novelty of this narrative review is that it examines a large body of investigations that have proposed HPN exclusively in cancer patients. The strength of this review is in the work of summarizing an extensive amount of data on this topic and balancing and presenting most of the available information in a series of practical tips.

Some limitations should be acknowledged. First, this is a narrative review and not a systematic review. Second, due to lack of evidence about the monitoring practice of cancer patients on HPN, the practical tips discussed in this review rely mainly on expert opinion. Finally, the monitoring process described is tailored on the needs of cancer patients and may be not applicable for other patients on HPN, such as those receiving this therapy for non-oncological diseases or for long-term periods.

## 17. Conclusions

Nowadays, we can expect an increased number of cancer patients in need of HPN, because more patients have access to effective anticancer treatments which are able to transform the trajectory of their disease in a chronic clinical condition. In these patients, nutritional therapy is crucial for survival and QoL maintenance. Thus, it is relevant to be prepared on how these patients should be best monitored and managed.

HPN is a complex therapy and its safety is mainly assured if cancer patients are carefully monitored over time. The process of monitoring the response to HPN requires the expertise of a multiprofessional, interdisciplinary team skilled in the management of all aspects of this treatment, such as choosing an appropriate CVAD, developing a PN prescription, diagnosing and treating complications, and weaning treatment when indicated.

The successful monitoring of cancer patients on HPN relies on the teamwork among physicians and dietitians at the discharging hospital, pharmacists, and home nurses, all with expertise in HPN management. In addition, the role of patients, caregivers, and general practitioners is pivotal; particularly, their active participation in carrying out PN procedures at home and in adhering to the planned monitoring program and in-hospital scheduled visits is crucial. Monitoring these patients, regardless of whether receiving or not anticancer treatments, should rely on affordable indicators evaluated on a regular and scheduled basis. The monitoring program should include tools evaluating nutritional and functional status, body composition, complications, and QoL.

In summary, such process of monitoring the response to HPN could be clinically useful to better guide healthcare providers in the nutritional care of cancer patients on HPN.

## Figures and Tables

**Figure 1 healthcare-08-00183-f001:**
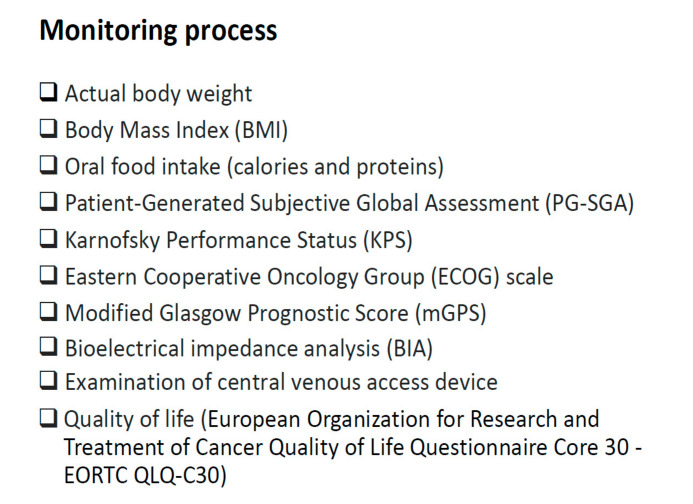
Parameters and indicators to monitor the response to Home Parenteral Nutrition in cancer patients.

**Figure 2 healthcare-08-00183-f002:**
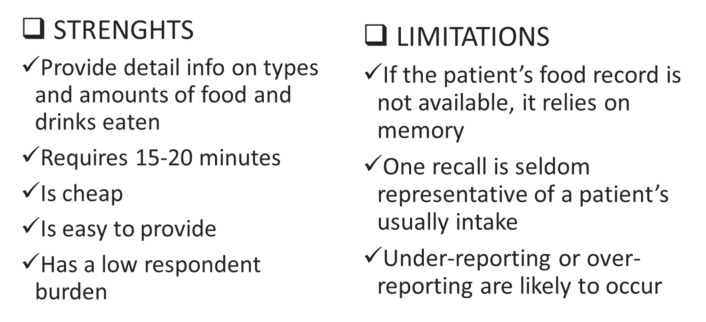
Oral food intake by 24-h food recall.

**Figure 3 healthcare-08-00183-f003:**
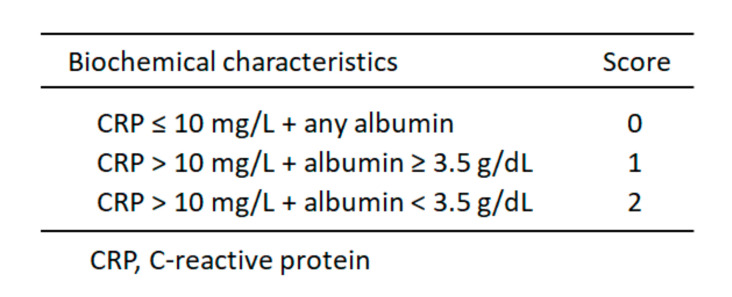
Modified Glasgow Prognostic Score (mGPS).

**Figure 4 healthcare-08-00183-f004:**
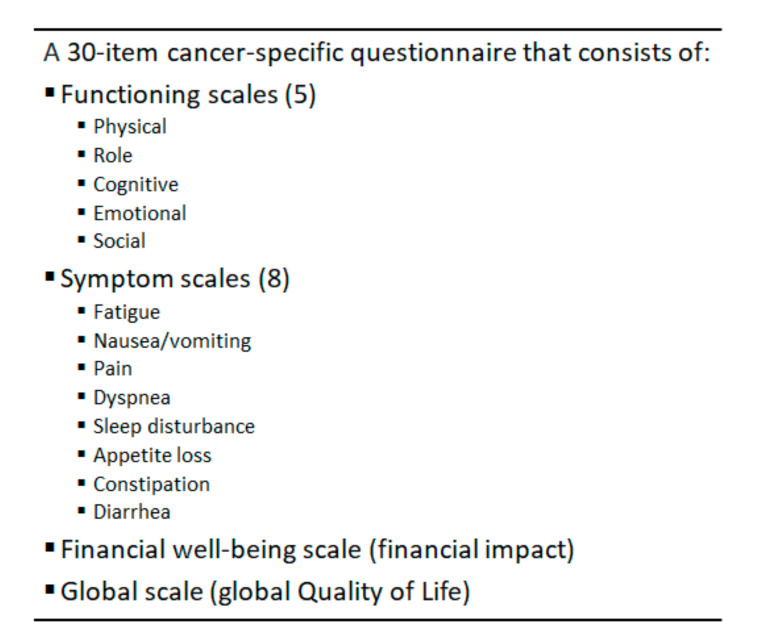
EORTC QLQ-C30: European Organization for Research and Treatment of Cancer Quality of Life Questionnaire Core 30.

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
