# Peer review of "Monitoring Response to Home Parenteral Nutrition in Adult Cancer Patients"

_healthcare, 2020, doi:10.3390/healthcare8020183_

Round 1

Reviewer 1 Report

The review presents the current knowledge on monitoring the response to HPN  in cancer patients.

The review of literature is well and extensively done however it is a narrative review

The authors propose easy implementable and practical protocol to monitor the response.  They present their procedure of monitoring as well as expiriences and clinical results.They also point its limitations of the procedure.

I have few remarks:

The review focuses only on adult patients so I woul suggest to include it in the title - ".....response to HPN in adult cancer patients"d prevalnence of HPN in adult cancer patients

In the section of CVAD lin 280-285, I would expolre the complication rate, I would suggest to present the types of possible complication with the numbers.

Author Response

  1. The review focuses only on adult patients so I woul suggest to include it in the title - ".....response to HPN in adult cancer patients"d prevalnence of HPN in adult cancer patients.        R. Thank you for the suggestion. As suggested by the Reviewer, we have added the word ‘Adult’ to the title.

  1. In the section of CVAD lin 280-285, I would expolre the complication rate, I would suggest to present the types of possible complication with the numbers.                                R. Thank you for the suggestion. As suggested by the Reviewer, we have discussed in Section n. 13 the incidence rates of catheter-related complications (page 8, lines 322-326).

Reviewer 2 Report

The manuscript by Cotogni et al, entitled Monitoring Response to Home Parenteral Nutrition in Cancer Patients is very interesting and reviews a large body of investigations that proposed home parenteral nutrition especially in cancer subjects. The authors are commended for the monumental work of summarizing extensive amount of data on this topic. The manuscript is clear, well written. The authors did an outstanding job in balancing and putting most of the available information in a series of fantastic flow. However, adding few schematic sketches would improve in general to audience/ scientific community and help to understand in a better way.

Author Response

  1. However, adding few schematic sketches would improve in general to audience/ scientific community and help to understand in a better way.                                             R. Thank you for the suggestion. As suggested by the Reviewer, we have added a few schematic sketches.

A native English speaker has revised the manuscript to improve the English language.

Reviewer 3 Report

The aim of the review is to describe the role and relevance of monitoring response to home parenteral nutrition in cancer patients.

Thank you for a nice manuscript, I only have a few comments:

What is new?

I miss awareness of refeeding syndrome (administration of HPN and monitoring). How to handle the risk of refeeding syndrome?

What are the strengths of this review?

Author Response

Point 1. What is new?

R. Thank you for the comment. We have discussed in Section n. 16 the novelty of this review (page 10, lines 393-398).

Point 2. I miss awareness of refeeding syndrome (administration of HPN and monitoring). How to handle the risk of refeeding syndrome?

R. Thank you for the suggestion. As suggested by the Reviewer, we have discussed in Section n. 5 how to handle the risk of refeeding syndrome (page 3, lines 114-118).

Point 3. What are the strengths of this review?

R. Thank you for the comment. We have discussed in Section n. 16 the strength of this review (page 10, lines 398-400).

A native English speaker has revised the manuscript to improve the English language.

Round 2

Reviewer 2 Report

Accept 

This manuscript is a resubmission of an earlier submission. The following is a list of the peer review reports and author responses from that submission.